# Insights into the Role of Telomeres in Human Embryological Parameters. Opinions Regarding IVF

**DOI:** 10.3390/jdb9040049

**Published:** 2021-11-13

**Authors:** George Anifandis, Maria Samara, Mara Simopoulou, Christina I. Messini, Katerina Chatzimeletiou, Eleni Thodou, Alexandros Daponte, Ioannis Georgiou

**Affiliations:** 1Department of Obstetrics and Gynecology, Faculty of Medicine, School of Health Sciences, University of Thessaly, 41222 Larisa, Greece; pireaschristina@gmail.com (C.I.M.); daponte@med.uth.gr (A.D.); 2Department of Anatomical Pathology, Faculty of Medicine, School of Health Sciences, University of Thessaly, 41222 Larisa, Greece; msamar@uth.gr (M.S.); eleni.thodou@gmail.com (E.T.); 3Laboratory of Physiology, Medical School, National and Kapodistrian University of Athens, 11527 Athens, Greece; marasimopoulou@hotmail.com; 4Unit for Human Reproduction, 1st Department of Obstetrics and Gynecology, Faculty of Medicine, School of Health Sciences, Aristotelian University of Thessaloniki, 56403 Thessaloniki, Greece; katerinachatzime@hotmail.com; 5Laboratory of Medical Genetics, Faculty of Medicine, School of Health Sciences, University of Ioannina, 45110 Ioannina, Greece; igeorgio@uoi.gr

**Keywords:** telomeres, gametes, embryos, assisted reproduction

## Abstract

Telomeres promote genome integrity by protecting chromosome ends from the activation of the DNA damage response and protecting chromosomes from the loss of coding sequences due to the end replication problem. Telomere length (TL) is progressively shortened as age progresses, thus resulting in cellular senescence. Therefore, TL is in strong adverse linear correlation with aging. Mounting evidence supports the notion that telomeres and male/female infertility are in a close relationship, posing the biology of telomeres as a hot topic in the era of human-assisted reproduction. Specifically, the length of sperm telomeres is gradually increasing as men get older, while the telomere length of the oocytes seems not to follow similar patterns with that of sperm. Nonetheless, the telomere length of the embryos during the cleavage stages seems to have a paternal origin, but the telomere length can be further extended by telomerase activity during the blastocyst stage. The latter has been proposed as a new molecular biomarker with strong predictive value regarding male infertility. As far as the role of telomeres in assisted reproduction, the data is limited but the length of telomeres in both gametes seems to be affected mainly by the cause of infertility rather than the assisted reproductive therapy (ART) procedure itself. The present review aims to shed more light into the role of telomeres in human embryological parameters, including gametes and embryos and also presents opinions regarding the association between telomeres and in vitro fertilization (IVF).

## 1. Introduction

Aging is an irreversible process accompanied by cellular senescence. This physiological process has been further characterized by a decline in reproductive potential. Millions of couples worldwide are affected by infertility, due to aging factors deteriorating their reproductive dynamics. The negative impact of reproductive aging has been in the spotlight of research for decades while emphasis is given to the rapidly declining reproductive potential of women older than the age of 35 [1,2]. It is evident that there is in fact an etiological relation between aging and reproduction. A known parameter employed to determine aging is considered to be the length of the telomeres. Telomeres progressively shorten as age progresses, resulting in the activation of a permanent state of cell cycle arrest known as senescence [3]. Therefore, the measurement of TL is considered a useful and valuable tool for predicting lifespan, as well as the reproductive biological age and potential [4]. Although the process of aging cannot be overridden, it seems that anti-aging factors, such as telomerase, have been proposed as potent candidates for the treatment of age-related diseases [5]. Telomerase activity (TA) has been associated with telomere lengthening as well as telomere length stabilization [6]. However, it is inactivated in somatic cells where telomeres shorten in each cell division. Telomerase re-activation in somatic cells occurs as a consequence of activating mutations or chromosomal rearrangements during tumorigenesis.

Accumulating data have proposed the association between TL and reproductive potential since aging is highly correlated with reproductive efficiency. Although TL is gradually undergoing erosion, the telomeres of the germ cells appear to follow a unique pattern allowing for maintaining the TL in each cell division. Sperm telomeres of older men seem to be longer in comparison to those from younger men [7]. Nevertheless, shorter sperm telomere length (STL) has been associated with various male infertility issues [8,9], indicating that the assessment of STL could be employed as a potential and valuable biomarker of male infertility in the clinical set-up. On the other hand, the length of the oocytes’ telomeres appears to be longer than in the spermatozoa [10]. The telomeres of the embryos are longer in the blastocyst stage in comparison to the early stages of the development [11], implying that the telomeres during the fertilization and early embryonic developmental stages are inherited by the gametes. The TL reset at the blastocyst stage seems to play a crucial role in establishing longer telomeres in the embryo [12]. Thus far, mounting evidence has correlated the embryo quality and IVF with TL. Specifically, aneuploid embryos demonstrated shorter telomeres, which has been inevitably associated with lower pregnancy rates [13]. In this regard, this aneuploidy is also frequently detected during cancer development and initiated by critically short telomeres. Whether aneuploidy is driven by telomere shortening or by chromosome segregation during meiotic division remains to be clarified, despite the notion that short telomeres may not successfully complete the meiotic divisions and subsequently result in various aneuploidies.

The aim of the present review is to shed light on the recent emerging role of telomeres on embryological parameters and to discuss the recent advances that have surfaced regarding the role of TL in ART.

## 2. The Role of Telomeres

Telomeres are specific repeated heterochromatic DNA sequences (TTAGGG)n that “seal” the ends of eukaryotic chromosomes ascertaining genomic integrity while preserving all the essential biological functions. Telomere length is established in germ cells during development by the activity of telomerase, determining the mitotic clock of each cell. In somatic cells, telomerase is inactivated leading to progressive telomere shortening, approximately 50–200 bp following in each cell division until they reach a critical length where senescence is activated. Besides progressive telomere shortening due to the end replication problem, telomere shortening is intensified by many genetic, environmental and lifestyle factors [14,15,16], therefore accounting for the high variability of telomere length observed in humans. The maintenance of telomere integrity is conducted by a complicated structure that involves the enzyme, telomerase and a core of proteins, named shelterin complex [17]. The role of this well-designed architecture has been elucidated to an extensive degree. Telomerase enzyme consists of two essential subunits, known as human telomerase reverse transcriptase (hTERC) human RNA component (hTR) [18]. As ribonucleoprotein reverse transcriptase molecule binds with high affinity to telomeres and adds TTAGGG sequences in order to extend the 3′ overhang site [6]. On the other hand, the shelterin complex consists of six subunits, including the telomere repeat binding factors, TRF1 and TRF2, protection of telomeres protein 1 (POT1), the TRF1-interacting nuclear protein 2 (TIN2), the tripeptidyl peptidase 1 (TPP1) and the repressor activator protein (Rap1). All of them in combination with other shelterin accessory factors are interacting and finally form a nucleoprotein complex that plays a structural protective role for the telomeres. It becomes evident that mutations in sheltering proteins such as TRF1, TRF2 or POT1 also lead to telomere shortening by perturbing chromosome end protection [19,20].

Telomeres are of both maternal and paternal origin and are inherited during fertilization. Although telomere inheritance has not been fully elucidated yet, it appears that telomere length constitutes the main factor for the onset of age-related diseases and is therefore correlated with life expectancy. Parents with long telomere length inherit long telomeres to their offspring which in turn provides a mitigated risk for age-related diseases [21]. Current evidence demonstrates that the telomere length of somatic cells is significantly reduced in comparison to the germ cells [15]. A number of telomere-based therapies have been introduced in order to mitigate telomere shortening suggesting the use of several activators, such as TA-65 (known as cycloastragenol), which has been shown to be effective in telomere length sustenance [22]. Furthermore, similar results have been reported in mice [23]. Despite the fact that telomerase constitutes a key component for maintaining telomere length it has been noted that the telomerase-dependent telomere lengthening is not the only mechanism safeguarding telomere maintenance. Alternative lengthening of telomeres (ALT) has been described during normal conditions but is not yet fully elucidated [24]. It appears that ALT leads to the formation of higher telomere lengths in comparison to the lengths observed following telomerase activity [25]. Break-induced replication (or Break-induced telomere synthesis) is the mechanism behind telomere elongation by ALT pathway [26,27,28,29]. This mechanism is under the control of several endogenous and exogenous factors that affect the telomeres’ length. Despite the presence of several natural anti-aging molecules, such as nutrients and other compounds that enhance telomerase activity, the senescence-aging phenomenon cannot be bypassed unless inactivating mutations accumulate in proteins involved in cell cycle regulation, such as p53 and RB. The latter represents the basis of tumorigenesis in aged people. On the other hand, there are a number of exogenous environmental factors that following exposure may enable a decrease in telomerase activity, promoting a senescence-aging outcome. Cigarette smoking and pesticides have been reported to affect gamete biology [30,31,32], which consequently may influence either telomerase activity or the telomere-shortening rate. In a similar way, endocrine disruptors have been proposed as the culprits for reproductive deregulation [33,34] and therefore telomere length issues. To summarize, even when chronological aging has not yet unfolded, genetic- and epigenetic-related factors may accelerate the senescence-aging phenomenon. As far as the endogenous factors aspect is concerned, normal diet and the supplementation of basic nutrients not only preserve a normal body mass index (BMI), which is correlated with male and female infertility cases [35], but further reduces the telomere-shortening rate [36].

Along these lines of genetic- and epigenetic-related factors that exert an impact on telomere length, it is worth highlighting the prolonged role of culture of gametes and pre-implantation embryos. It is known that extended culture of embryos may be associated with the developmental competence of the embryos, as well as with epigenetic effects on the developing embryo [37,38]. As suggested, the telomere length is more related to the infertility status of the patient rather than the ART process [39]. Whether extended embryo culture exerts a negative impact on embryo telomeres remains to be elucidated, nonetheless, several points could be raised. For instance, the telomeric repeat-containing RNAs, known as TERRA, have been positively associated with embryo quality in in vitro fertilization/intracytoplasmatic sperm injection—embryo transfer (IVF/ICSI-ET) treatments [40]. Moreover, pregnancy success rates in the context of ART have been associated with shorter telomere length in newborns [41]. In that study, a number of factors have been investigated, including BMI, smoking status, alcohol and stress. Although not yet documented, it can be hypothesized that telomere length might have been affected by culture embryo extension during the ART process, which epigenetically and indirectly impacts pregnancy success rates. 

Finally, the reverse transcriptase (RT) enzyme telomerase is organized in a complex ribonucleoprotein (RNP), with a specific RNA template that supports telomere maintenance and elongation [42]. Although the two RNA templates (telomeric and retroelement) are different and they do not overlap in the genome, they have in common an adaptive response to chromosome stability. Especially in the gametes and early embryos when the overall methylation changes are marked and the methylation control over retroelements, retrotransposons and RTs is reinstated, retroelement transcripts are overexpressed and present in the oocyte just before fertilization [43,44]. Specifically, in Drososphila, retrotransposon reverse transcriptase has been suggested to compensate for the absence of telomerase in the genome or when telomerase is inadequately expressed [45]. On the other hand, inhibition of RTs in the ovaries, the testis and early embryos of experimental animals, compromises the gamete maturation process and drives to embryo collapse before the formation of blastocysts [46,47].

## 3. Telomeres and Spermatozoa

As it is known, sperm DNA should be transferred to the zygote as intact as possible, since it constitutes the half genetic material of the zygote. Apart from the genes, it seems that sperm DNA telomeres also contribute to the zygote and consequently sperm telomeres are of crucial significance. Over the last decade, sperm telomeres have been strongly associated with male infertility [48], implying that the telomeres of the spermatozoa are key players in defining telomeres of the offspring. Mature spermatozoa appear to contain longer telomeres (sperm telomere length, STL) in comparison to the telomeres of the somatic cells (leucocyte telomere length, LTL). Employing sophisticated methods for the measurement of STLs, such as qRT-PCR or quantitative fluorescence in situ hybridization (Q-FISH) it became clear that sperm telomere length was approximately 10 to 20 Kb [9,16]. Elevated telomerase expression has been detected during the early phases of spermatogenesis, while this expression is declining during spermiogenesis representing the late phase of spermatogenesis [49,50]. These findings imply that spermatocytes maintain their telomere length, while spermatids and mature spermatozoa present with shorter telomere lengths. An additional explanation for this intra-variability expression of telomerase could be attributed to the process of spermatogenesis/spermiogenesis. In other words, the late phases of spermatogenesis involve a number of complicated chromosome re-organization procedures. Telomerase preferentially is highly expressed in the early phases of spermatogenesis, stabilizing the length of the telomeres during its initial stages [51]. Nevertheless, it is evident that STL is increasing as paternal age is increasing while, paradoxically, LTLs undergo age-related erosion. Moreover, older paternal age at conception predicts longer TL in offspring [52]. The above documentation confirms the telomere length heritability and implies longer telomeres for future progeny [53]. In an effort to address this discrepancy between older and younger men, the fact that spermatogenesis in older men seems to exhibit different mitotic divisions compared to younger men could serve as the basis to interpret why older individuals present with longer telomeres. The question raised is in which paternal age telomere length is starting to increase. The high STL heterogeneity may be attributed to the various environmental factors that gametes might have been exposed or to the variability that the ALT mechanism exhibits in each individual.

Despite the fact that the association between paternal age and TL in offspring is still considered a heated controversial topic, accumulating evidence has shed light on the possible relation between STLs and male fertility/infertility. Specifically, it has been demonstrated that infertile men present with shorter STLs compared to fertile men [54]. These results have been further supported by findings indicating that in oligospermic samples shorter STLs were detected when compared to normospermic samples. Additionally, sperm DNA fragmentation along with a percentage of aneuploidy was significantly higher in samples with shorter STLs in comparison to samples with longer STLs [55]. A relationship between deteriorated sperm parameters—including sperm motility and sperm count—with shorter STLs has been noticed, as well as a correlation between idiopathic male infertility and shorter STLs [15,54]. The evidence on STLs and male reproduction is accumulating, while the exact underlying mechanisms that interprate the effect STLs exert on the reproductive potential are currently in the spotlight of experimental studies. It is a well-established knowledge that protamination is a normal process during spermatogenesis, during which sperm protamines are gradually replaced by histones. Any disruption during the protamination process could result in abnormal spermatogenesis, which could inevitably lead to male infertility due to the increased events of apoptosis and the elevated genomic instability [56]. Abnormal protamination has been linked to shorter STLs which therefore compromise male fertility [55,57]. Nevertheless, according to Hemann and colleagues [58] an interesting surveillance mechanism has been proposed, where spermatocytes with shorter TLs do not complete the process of spermatogenesis and selectively undergo apoptosis in order to selectively discard abnormal spermatozoa or spermatozoa with shorter TLs. Another aspect that has been explored in literature in an effort to interpret the aforementioned results is the oxidative stress parameter. Numerous studies have demonstrated that increased oxidative stress has been related to abnormal sperm parameters, including sperm DNA fragmentation [59]. Any alteration in the interplay between protamines and histones may possibly increase the susceptibility of sperm DNA to factors such as ROS, which in turn promotes an enhanced percentage of sperm DNA fragmentation [59]. The above situation inevitably leads to the shortening of STLs [48], while the inverse correlation between STLs and ROS has been further confirmed in literature [60]. Interestingly, while normal and mild oxidative stress is associated with longer STL, severe oxidative stress is related to shorter STL [61]. Therefore, it remains vague whether STLs poses a risk factor for male infertility or whether it solely constitutes an additional parameter for evaluating male infertility. The TERT rs2736100 was inversely associated with male infertility risk, whereas TEP1 rs1713449 was positively associated with risk of male infertility [62], constituting that STLs could indeed pose as a risk factor for male infertility.

## 4. Telomeres and Follicles/Oocytes

Investigating the role of telomeres in female gametes and female infertility, it becomes evident that striking differences compared to spermatogenesis and STLs exist. Moreover, even though data is limiting it seems that telomere length in female gametes follow a different pattern compared to sperm. Nonetheless, it is known that telomerase activity is relatively high during the early stages of oogenesis while it further declines during maturation, at the late phases of oogenesis [11], indicating that oocyte telomeres (OTs) are longer at the early stages and shorter at the late stages of oogenesis, determining the final oocyte telomere length (OTL). Various internal and external (environmental) factors may further trigger the shortening of OTs during oogenesis. A major internal factor that contributes to oocyte aging and extended meiotic arrest is advanced maternal age. The gradual decrease of the quantity, as well as the quality of the oocytes observed due to the female reproductive aging, is considered granted. During this transition, it has been reported that advanced maternal age could be associated with relatively shorter telomeres compared to the telomeres in younger reproductive ages [63]. It is possible that the advanced reproductive age is the trigger agent for the shortening of the telomeres of the oocytes and truncated telomeres decrease oocyte quality during advanced maternal age. Hitherto, there is a notion that extended meiotic arrest enables the accumulation of ROS, which in turn results in oocyte telomere attrition due to the chronic exposure of telomeres to ROS [64].

Apart from the advanced maternal age, another internal factor that exerts an impact on the fate of TL is the BMI of women. As indicated in the literature, age and BMI are closely related factors responsible for the onset of several fertility issues [35]. Overweight or obese women seem to exhibit a reduced rate of achieving pregnancy in comparison to normal-weight women due to the increased levels of ROS. It is, therefore, evident that, depending on ROS levels, it could lead to either telomere shortening or lengthening. Since increased BMI is associated with several female fertility issues, such as polycystic ovarian syndrome (PCOS), where specific genetic factors are involved in the pathogenesis of this syndrome, TLs seem to play role in the onset of this disorder as well. Women with PCOS appear to correlate with shorter telomeres in comparison to fertile women [65]. The correlation between TLs and the methylation—de-methylation cycle has been not elucidated to an extent [66]. An explanation for this discrepancy could be the proliferative function of granulose cells during ovarian stimulation which in turn indicates an increased TA. While the aspect of TLs in PCOS women remains controversial, the origin of the cells examined for the TLs appears to be the major factor for evaluating TLs.

Furthermore, exogenous and mainly environmental factors may indirectly affect TL. Pollution and extended exposure to toxic agents, namely pesticides or herbicides that promote endocrine disorders (EDs), seems to increase the levels of ROS, which subsequently triggers the acceleration of reproductive age, telomere erosion and finally compromises oocyte performance. Oocyte quality is a prerequisite for a successful fertilization and embryo development and is essential for ascertaining female fertility. To date, numerous studies have been conducted in order to determine any potential relation between TLs and female fertility/infertility. Specifically, women undergoing IVF-ET treatments—regardless of their etiology—seem to present with shorter TLs in comparison to healthy controls [67]. It appears that the etiology of infertility constitutes the trigger for telomere erosion, which results in sub-fertility. This notion has been further supported by the assessment of LTLs in women who have at least one live birth compared to women who have solely experienced a biochemical pregnancy, spontaneous abortion or miscarriage. Significantly longer TLs have been documented in the first group of women in comparison to the second group [68]. Premature ovarian failure (POF), a disorder almost synonymous with infertility has also been linked to shorter TLs [69]. The pathological cessation of ovarian function coupled with the fact that the ovaries fail to produce and secrete adequate levels of estrogens to function properly prior to the age of 40 appears to be related to TLs. Telomere length seems to define the fate of the ovarian state in this cohort of patients. The lack of TA due to the absence of estrogens has been suggested as a plausible explanation for the above correlation between POF and TLs [70].

In addition to TLs, another biological and molecular marker of aging is DNA methylation. However, the correlation between TLs and the methylation—de-methylation cycle of the gametes, including the oocytes remains to be elucidated. Aberrant DNA methylation, similar to telomere shortening, seems to present with significant correlations with aging and the onset of age-related diseases [71]. In one study, a positive correlation between DNA methylation and TLs was demonstrated [72]. On the other hand, aberrant methylation leads to various disorders, but further studies are needed in the context of telomeric chromatin. It seems that epigenetic signatures established during the early stages of gametogenesis may contribute more extensively to determining the TL and not that TL defines the fate of the methylation state [73]. Additional studies will uncover mechanistic clues. 

## 5. Telomeres and Pre-Implantation Embryo Development

Infertility has been associated with shorter telomeres nonetheless the question posed is whether TL could be implemented as a tool for predicting embryo quality and embryo developmental potency. As evidenced by extensive literature research, data is very limited. Interestingly, the evolutionary notion that embryos originating from the fusion of normal gametes exhibit increased probability for a longer lifespan and decreased risk of developing diseases, might have been based on the concept of telomeres length. Telomerase appears to maintain a relatively low activity during the fertilization process and the initial stages of embryo division, while at the morula and the blastocyst developmental stage TA is gradually increasing [11], reaching the levels of the activity described in germ cells [74]. This cycle observed in the function of telomerase appears to have inverse similarities with the methylation—de-methylation cycle. TA is low during the fertilization stage, while during the same stage methylation is highly active. Similarly, TA presents high during the blastocyst stage, while methylation has been detected to be low during the same developmental stage (Figure 1). Germ cells exhibit high methylation levels, while TA varies between initially high and late low levels during the spermatogenesis process (Figure 1). As in the methylation—de-methylation cycle which establishes the methylation pattern of the offspring, it seems that telomerase determines new TLs during embryo development and especially during the blastocyst stage. This concept of TL reset theory has been proposed in the literature [12]. It is possible that both of these two active anti- parallel processes are intertwined in order to achieve the optimal physiological and genetic developmental potency. This assumption may be valid taking into consideration that methylation levels are high during the fertilization process, which accounts for silencing gene expression, including the expression of telomerase. Therefore, it becomes evident that the TLs of the embryos are inherited by the TLs of the gametes. It is very possible that gametes with long telomeres will result in embryos with long telomeres. Following in the same line, embryos with short telomeres might have been originated from gametes presenting with short telomeres. At this point, the question raised is whether both gametes contribute equally to determining the length of telomeres of the embryos. There is a notion that TL of the embryos is determined by the STL, as it has been documented in animal models [75], while the correlation between STL and TL of the embryos abides by an age-dependent pattern as it has been described above. The telomeres of the oocytes are longer than those of the spermatozoa [10], nonetheless as the reproductive age of women increases, OTL is declining. On the contrary, as the reproductive age of men is increasing, STL is paradoxically increasing. Therefore, it can be assumed that the optimal spermatozoon-oocyte match for ensuring long telomeres and extended lifespan might involve a spermatozoon from a male individual of an advanced age along with an oocyte from a young female. Although the differences in the length of telomeres between oocyte and spermatozoon at the time of fertilization, it appears that the embryo inherits the longer telomeres. The re-establishment of the telomeres during blastulation according to the TL reset theory could potentially determine the TL of the embryos. However, telomere length has a limit which depends on the amount of shelterin protein that can protect the DNA structure. In that case, a protein called TZAP binds to DNA ends and trimmer telomeres to the length that can be protected by the levels of shelterin proteins [76]. Nevertheless, intra-variability of TL between embryos originating from the same parents has also been documented [77]. This adds another level of complexity to our efforts to unravel the underlying mechanisms involved in this phenomenon. 

The genetic and cytoplasmatic contribution of the oocyte to the early stages of embryo development is well documented [78]. In case of a prolonged prophase during the first cycle, segregation errors which may occur during meiosis in the oocytes have been observed [79]. Errors during chromosome segregation may in turn result in genetic malfunction of the telomeres and subsequently, embryos may inherit shorter telomeres. Whether sperm DNA can repair the shorter telomeres of the oocyte or the other way round merits a deeper level of investigation. During fertilization and sister chromatid exchange, female pronuclei telomeres can repair male pronuclei telomeres when they exhibit short telomeres in order to maintain genetic stability [80]. The reset length theory, which takes place during blastulation as mentioned earlier, seems to support the above assumption, indicating that the embryo has acquired the appropriate mechanisms to ensure longer telomeres. On the other hand, the threshold of either OTL or STL contributing to a compromised fertilization and embryo development has yet to be established. It is of paramount significance to determine the fine line that characterizes abnormal cases of OTL and STL since they pose a risk for total reproductive failure. The ALT pathway seems to participate during the early stages of development prior to the morula-blastocyst stage [77,81]. It has yet to be determined whether ALT is an alternative mechanism supporting telomere lengthening during the cleavage stage of embryos, albeit it has been proposed to contribute to the TL heterogeneity of the embryos [80] during the first embryo divisions. ALT function could potentially serve as a mechanism to maintain telomere stability regardless of the length and methylation status observed in the telomeres of each gamete. Lastly, this mechanism seems to cease prior to the formation of the morula and blastocyst, since telomerase and TL reset are the main regulatory components at this stage. Additional data regarding the role of telomeres in embryo quality and development have been showcased in studies focusing on the evaluation of TL in couples undergoing infertility treatment in the context of assisted reproductive technologies. This interesting matter is further discussed in the next section.

## 6. Opinions Regarding Telomeres and ART

Considering that TL may exert a similar impact on ART cycles as the cyclical methylation/de-methylation process, the notion that TL should be meticulously examined under the prism of ART demands to be further discussed. Besides, shorter TL has been proposed to correlate with both male and female infertility, as described in the previous sections. The process of various assisted reproductive techniques may compromise DNA integrity and function of both gametes [32]. For instance, extended embryo culture has been associated with imprinting disorders owing to an abnormal methylation—de-methylation cycle [37]. ICSI, constituting an invasive insemination method, has been associated with increased risk of congenital defects. Taking into consideration the above correlations, the possible relation between telomeres and ART cannot be excluded. However, data is limited, due to the ethical concerns raised by employing human embryos for research purposes.

Sperm preparation either employing the swim-up or the density gradient method involves isolation of a sperm population with increased motility, longer telomeres and decreased percentage of DNA fragmentation [82]. As suggested also, the prepared sperm aliquot is enriched with spermatozoa displaying long telomeres [83]. On the contrary, it has been observed an association between poor semen quality and shorter TL [57,84]. The physiological process of spermatozoa moving through the cervix while preparing for the acrosome reaction, is similar to sperm preparation process. Therefore, someone could speculate that the majority of sperm ejaculate during the physiological process of fertilization is enriched with spermatozoa characterized by long telomeres. The mechanism allowing spermatozoa with short telomeres to achieve fertilization seems to depend on the female micro-environment and the interaction between the female reproductive tract and spermatozoa. Hitherto, extended embryo culture has been suggested to result in an increased risk of imprinting disorders [37], however no evidence elaborating on whether extended embryo culture may affect TLs has been reported. Given the fact that extended embryo culture may pose a risk to the methylation process, it could be assumed that extended embryo culture may further exert an impact on TA or even on TL. STL has been correlated with good quality embryos [85], indicating that the selection of spermatozoa with long telomeres is crucial for development of good quality embryos, and potentially for successful IVF outcome [83]. IVF failure and increased risk for recurrent miscarriage have been associated with embryo aneuploidy, while short telomeres are considered to be the leading cause of aneuploidy and delayed embryo development [13]. Recently, despite the fact that STL was not found to be correlated with sperm and other embryological parameters, it has been associated with IVF success rates [86]. Therefore, what appears to stand as a valid hypothesis is that pregnancy success rates, either referring to natural conception or following an IVF attempt, depend on the TL since the latter seems to reflect on embryo quality. This assumption is supported further by the finding that longer STLs could be associated with significantly higher natural pregnancy rates, while shorter STLs were related to lower natural pregnancy rates [9].

Despite the fact that BMI is playing a major role in the reproductive outcome, it seems that BMI exerts a strong effect on the length of the telomeres, posing both parameters as crucial agents for the IVF outcome [87]. On the contrary, STL in donor samples does not hold the potential to predict ICSI outcome [88], since the quality of sperm donors are lacking any diseases and foremost the age of the sperm donors is relatively low. Altered gene expression of shelter in and telomere-associated proteins such as the TERB1, TERB2 and MAJIN genes, has been observed in men with non-obstructive azoospermia [89], indirectly indicating that the expression and the function of the telomere complex is essential for the normal function of spermatogenesis, as well as for the production of spermatozoa with normal parameters. The telomeres of women in non-complicated pregnancies either physically or through assisted reproductive technologies appear to be longer in comparison to patients with complicated pregnancies, such as cases of intrauterine growth restriction (IUGR) [90]. Nonetheless according to the conclusions of this study TL was not associated with the pregnancy outcome. The existing correlations between TL and specific reproductive parameters and especially the reproductive outcome warrant further investigation, since TL or TA could be employed either for predicting the IVF outcome or as a biomarker of male and female infertility [91]. Taking into consideration that the evaluation of TL is a relatively accessible and affordable technique, it may fit the criteria to be introduced as a tool for predicting sperm and embryo quality, as well as an additional prognostic marker for pregnancy success.

## 7. Conclusions and Perspectives

The limited available data regarding the relationship between telomeres and fertility/infertility showcase the linear correlation between shorter telomere length and human reproduction. The mounting evidence supports the role of telomere evaluation as a biomarker of prognostic value indicative of a successful outcome in the ART context. Although STL has been associated with several sperm parameters, it becomes evident that employing STL as a unique biomarker for predicting sperm and embryo quality may lack feasibility, since quantifying the TL in each spermatozoon that will potentially fertilize one oocyte cannot be performed. On the contrary, the evaluation of TL in embryos may represent a valid option since performing a biopsy at the blastocyst stage stands as a straightforward procedure of clinical routine practice status in the context of pre-implantation genetic testing for aneuploidy (PGT-A), while assessment of TL in a blastocyst may predict not only the implantation and ongoing pregnancy potential, but further the lifespan of the newborns to be.

TL has been characterized as a marker of cellular aging and reproductive aging. The TL of leucocytes is not correlated with those of sperm or oocytes. LTs are shorter as age increases, while STLs are paradoxically longer as age progresses. The telomeres of embryos in the context of ART are considered to reset their length at the morula and blastocyst stage, while at the early stages of development telomeres appear to be shorter. It is clear that the measurement of TL in reproductive cells may enhance the predictive power and may improve the clinical strategy to manage infertility. Moreover, the measurement of TL may fill the knowledge gap regarding promising biomarkers in diagnosis and addressing male and female infertility. To the best of our knowledge, data published hitherto justifies the further research that is required in order to elucidate the exact role of telomeres in human-assisted reproduction.

## Figures and Tables

**Figure 1 jdb-09-00049-f001:**
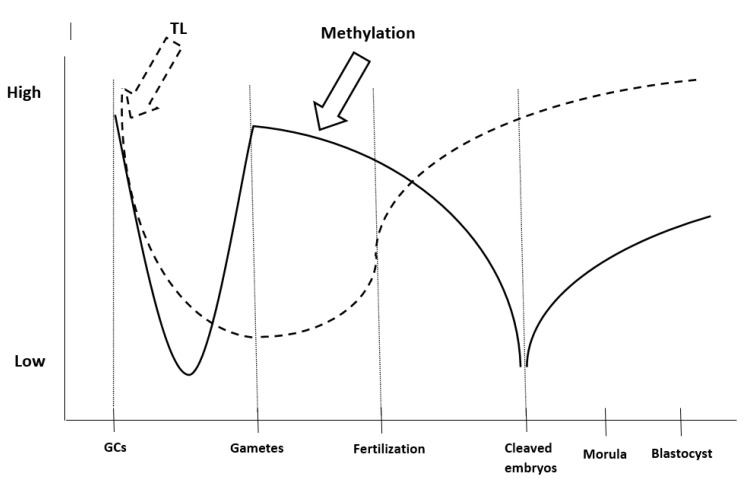
Telomere length reprogramming and methylation—de-methylation cycle during gametogenesis and early embryo development. The dot line represents the telomere shortening-lengthening, while the continuous line describes the methylation—de-methylation cycle taking part during the stages described above. GCs; Germ Cells, TL; Telomere Length.

## Data Availability

Not applicable.

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
