# Peer review of "Insights into the Role of Telomeres in Human Embryological Parameters. Opinions Regarding IVF"

_jdb, 2021, doi:10.3390/jdb9040049_

Round 1

Reviewer 1 Report

In this review article, George Anifandis and colleagues show extensive information about the role of telomeres in reproduction at the level of gametes (sperm and oocytes), fertilization/embryogenesis, with the perspective of using telomere length as a marker for IVF, and how ART could benefit from this information. Collectively, this review will be of interest to the audience of the Journal of Developmental Biology and more specific to the ones interested in telomeres and its relationship with the biology of reproduction. However, this reviewer thinks that some editing should be addressed which will be beneficial to improve the readability and overall quality of the review article.

General major comments:

  • This reviewer think that your review article will benefits from an extensive text editing, maybe with the help of a native English speaker. This editing will help to clarify connections between statements inside and between different sections which will be beneficial for the overall review organization.
  • Authors need to do some work with acronyms. They are not used or when defined, they are not consistently used across the text. I ask the author not to assume this review is only for people specialized in reproduction that knows the terminology, rather than for a bigger audience.
  • General information about telomeres and telomerase is confused and sometimes out of what is accepted in the field. Revision of those concepts together with appropriate and updated citations will be important to translate that information to the reproduction field.
  • I have spotted across the text multiples places with statements missing a reference. For a review article like this, references of the original articles where that knowledge was acquired should be added. For general mechanism, other updated reviews can be cited too.

Specific comments:

  • I think the authors should re-think some definitions about telomeres in different section and add appropriated references to them. For example, Telomeres promotes genome integrity by protecting chromosome ends from the activation of the DNA damage response which is referred to the end protection problem, and protecting chromosomes from the loss of coding sequences due to the end replication problem.
  • Lane 22. “Progressively shortening” should be changed for progressively shorten.
  • Lane 24. “Infertility is strongly associated with aging in both sexes”. I think this is not accurate with the information provided along the review concerning the positive correlation between age and telomere length in male (for example, in lane 360).
  • Lane 29. Telomere length in embryo seem to have parental origin. The authors state and I agreed that this is not clear, mostly because telomere length can be modified by telomerase activity early during embryogenesis. Maybe the author refers to the blastocyte stage before telomerase is more active. Clear separation and clarification of these possibilities should be key to understand what is inherited and then what telomere length is extended by telomerase.
  • Lane 47. Gradually and progressively are synonyms. Just use “telomeres progressively shorten as age progresses, resulting in the activation of a permanent state of cell cycle arrest known as senescence”
  • Lane 53-55. This statement is only valid in condition where telomerase is active, which during early development and to maintain telomeres in stem cells. However, it is inactivated in somatic cells where telomeres shorten in each cell division. Cells don’t adopt the mechanism of telomerase-mediated telomere elongation. Telomerase re-activation in somatic cells occurs as a consequence of activating mutations or chromosomal rearrangements during tumorigenesis.
  • Lane 70. Aneuploidy in embryos due to telomere shortening. This is interesting to happen in sperm cells, and is closely related with the fact that short unprotected telomeres can fuse with other chromosome free ends (either telomeres or internal sequences) leading to aneuploidy during chromosomal segregation in cancer development. It will be interesting to add this idea. This connection will be interesting to work on to complete the introduction section.
  • Lane 81-83. Should say “Telomere undergo progressive shortening following each cell division, until they reach a critical length where senescence is activated”. Shortening is not exactly intensified by age, is intensified by many endogenous and environmental factors which may accumulate as age progress. A reference to the work of Leonard Hayflick should be added.
  • Lane 85-86. Again, I don’t think equilibrium between shortening and lengthening is what is happening in all situations. That could be the case of early development or what is happening in stem cells, however somatic cells follow progressive shortening unless telomeres are damages and the associated repair led to transient lengthening. See your lanes 117-119, that is the general accepted model.
  • Lane 89-92. Telomere length is stablished during development (which determine the mitotic clock), then is progressive shortening leading to aging. The fact that telomeres shorten with each cell division does not involve that a mechanism to elongate them should be activated. This only happens in cancer cells due to telomerase re-activation that is required to maintain their high proliferative potential. Maybe the authors refer to specific cell lines or systems, but it should be stated.
  • Lane 93. References for shelterin complex is needed. Authors could add references from Titia de Lange lab.
  • Lane 95. When talking about telomerase, and maybe in other cases, I think the authors should acknowledge its discovery by adding the appropriate reference. Indeed, in this specific case, a nobel prize was awarded to Elizabeth Blackburn, Carol Greider and Jack Szostak for the discovery of how chromosomes are protected by telomeres and the enzyme telomerase. In general, references related to telomere section are missing.
  • Lane 99. POT1 is called “Protection of telomeres protein 1
  • Lane 103. Mutations in sheltering proteins such as TRF1, TRF2 or POT1 also lead to telomere shortening by perturbing chromosome end protection.
  • Lane 110-112. “ Generally, it is well stablished that…”. This type of statements required references.
  • Lane 125. ALT is activated in normal somatic cells. The references provided are referring to ALT mechanism rather than specific activation in normal cells.
  • Lane 126-129. Break-induced replication (or Break-induced telomere synthesis) is the mechanism behind telomere elongation by ALT pathway. Homologous recombination repair telomeres but does not lead to elongation. The ALT mechanism has been the focus of intensive research in the last years, the author needs to add appropriate and more recent bibliography.
  • Lane 134. Senescence can actually be bypassed by inactivating mutations in proteins involved in cell cycle regulation such as p53 and RB. This is actually the basis of tumorigenesis in aged people.
  • Lane 137. “which consequently may influence either telomerase activity or the telomere-shortening”. It is known that eternal factors have an influence, a reference is needed.
  • Lane 147. Are the authors talking about the length of culturing or also the composition of the culture media?
  • Lane 154. “the telomeric repeat-containing RNAs, known as TERRA, have been associated with embryo quality in IVF/ICSI-ET treatments”. It should be clear whether this association is positive or negative.
  • Lane 157. “In the present study, a number of factors have been investigated” What study the authors refers to. From the context, I may guess is reference 36.
  • Lane 159. It is not clearly documented, actually the authors of reference 36 said more work is needed in that way. I don´t think the statement of a relationship between telomere length and time of culture could be done from a reference where it wasn´t tested.
  • Lane 163-181. This section long and week on the information that supports it, only detected in Drosophila (it should specifically in Drosophila, rather than particularly in Drosophila). Maybe the authors have some unpublished observations that supports that. It could be either removed completely or the authors could only relocate the idea of RT activity on compensating for telomerase inactivation in the previous section.
  • Lane 184. “sperm DNA should be contributed to the zygote”. Maybe, should be transferred to the zygote.
  • Lane 192. Q-FISH is defined as Quantitative fluorescence in situ hybridization (Q-FISH).
  • Lane 193. “sperm telomere length was approximately 10 to 20 Kb”. What is the variation with age rather than telomerase? Chromosomal reorganizations are involved in telomerase re-activation in some cancer types, could be happening the same in the late phases of spermatogenesis.
  • Lane 205. “Moreover, it has been documented that telomeres of older men are longer than those of younger individuals”. Because is documented, a reference is needed.
  • Lane 210-212. “The question raised is what is the crucial cut-off point of paternal age at which point telomere length is starting to increase and remains to be elucidated taking into consideration the heterogeneity of STLs”. Need to be rephrased for a clearer understanding.
  • Lane 233-234. “Shorter STLs seems, therefore, to compromise male infertility, since abnormal protamination has been linked to shorter STLs” It should just say, “Abnormal protamination has been linked to shorter STLs which therefore compromise male fertility”.
  • Lane 243-244. “ROS, which in turn promotes an enhanced percentage of sperm DNA fragmentation”. A reference is needed here.
  • Lane 246-249. “Interestingly, ROS levels seem to play a crucial role in determining STLs Normal or mild oxidative stress, which could be expressed through the evaluation of ROS levels, is associated with longer STLs in comparison to severe oxidative stress which seems to be related to shorter STLs”. Hard to understand, it should say “Interestingly, while normal and mild oxidative stress is associated to longer STL, severe oxidative stress is related to shorter STL”.
  • Lane 252. rs2736100the. Remove “the” and authors should define better this polymorphism.
  • Lane 257-259. If the data is limited, then there is not a striking association. It should say, “even though data is limiting it seems that telomere length in female gametes follow a different pattern compared to sperm”
  • Lane 271-272. “whether truncated telomeres trigger oocyte quality during advanced maternal age, remains to be elucidated”. Authors should re-phrase or elaborate the possibilities on how this may happen.
  • Lane 279-280. “It is, therefore, evident that, ROS levels are crucial and essential for both telomeres lengthening and shortening”. It should be added that “depending on ROS levels, it could lead to either telomere shortening or lengthening”.
  • Lane 311-314. Is this referred to both gametes?
  • Lane 314. “…seems to present with significant correlation” change to “seems to correlate”
  • Lane 315-316. Here, reference 66 demonstrate a link between methylation and TL. It should be written what is the link (shortening, lengthening…). Besides that, this sentence seems to contradict another statement in lane 317, where authors said that “the impact of aberrant methylation to TLs cannot be excluded”. Please clarify if the relationship cannot be excluded or was demonstrated but further studies in the context of telomeric chromatin.
  • Lane 322. “…may raise further concerns”. I think additional studies “will uncover mechanistic clues”
  • Lane 333. Reaching the levels instead of resuming the levels.
  • Lane 339. “TA remain low throughout the oogenesis process”. In lanes 260-262, the author said the TA is high at early stages and low at the final phase of oogenesis. Please clarify.
  • Lane 343. As mentioned above, TL reset theory represent the accepted process to determine telomere length. What is happening before that state during fertilization and the initial stages of embryogenesis represents is obscure and the authors here have a great opportunity to fill some information in.
  • Lane 344-350. References are needed.
  • Lane 361-363. “…although the differences in the length of telomeres between oocyte and spermatozoon during fertilization”. It seems incomplete.
  • Lane 364. If telomeres are longer during fertilization process, I guess this will be the base for longer telomere extension and life span when telomerase gets activated during blastulation. However, telomere length has a limit which depend on the amount of shelterin protein that can protect the DNA structure. In that case, a protein called TZAP bind to DNA ends and trimmer telomeres to the length that can be protected by the levels of shelterin proteins.
  • Lane 372-373. “…is well documented”. Do the authors have references for this?
  • Lane 374-375. “…have been observed”. Reference needed.
  • Lane 377. DNA repairing polymerase? What mechanism do the authors refer to here? Is it a polymerase for HR-mediated repair mechanism?
  • Lane 383. Telomerase-mediated telomere elongation and HR-mediated repair at telomeres are not exclusive mechanisms and can be taking place at the same time, however the first one lead to elongation while the second one lead to telomere length heterogeneity as has been described in ALT cells. Telomere elongation independent of telomerase in ALT-related break-induced replication which may be taking place also (lanes 389-392). However, I’m agreed that further investigation will be needed to elucidate that.
  • Lane 390. It has to be determined whether…
  • Lane 394. What abnormalities the authors refer to? Length, methylation status…
  • Lane 397-400. “Additional data…”. References.
  • Lane 405. “TL has been proposed to correlate with both male and female infertility” At this point in the review, this statement should be clearer because the correlation between age and TL is positive in the case of male and negative in the case of females (also for lane 464). Therefore, the statement should be re-phrased to say that short telomeres correlate with infertility.
  • Lane 406. First time Assisted reproductive techniques (ART) is defined and acronym is not used.
  • Lane 408. “…abnormal methylation/de-methylation cycle”. Reference missing. This reference should be used for lanes 423-424.
  • Lane 415. Is it know the relationship between motility and TL, and how is the variability of TL in a sperm sample population?
  • Lanes 446. “…altered gene expression of the main gene telomere complex…”. It should say “ altered gene expression of shelterin and telomere-associated proteins such as…”
  • Lane 450. Change “characterize” for “with”
  • Lane 452. IUGR not defined.
  • Lane 472. PGT-A not defined.

Author Response

We thank the Reviewer for the corrections and the suggested comments. The below answers are given to criticism

Specific comments:

  • I think the authors should re-think some definitions about telomeres in different section and add appropriated references to them. For example, Telomeres promotes genome integrity by protecting chromosome ends from the activation of the DNA damage response which is referred to the end protection problem, and protecting chromosomes from the loss of coding sequences due to the end replication problem. The sentence has been deleted and replaced by the suggested one as the Reviewer proposed.
  • Lane 22. “Progressively shortening” should be changed for progressively shorten. It has been corrected
  • Lane 24. “Infertility is strongly associated with aging in both sexes”. I think this is not accurate with the information provided along the review concerning the positive correlation between age and telomere length in male (for example, in lane 360). The sentence has been deleted.
  • Lane 29. Telomere length in embryo seem to have parental origin. The authors state and I agreed that this is not clear, mostly because telomere length can be modified by telomerase activity early during embryogenesis. Maybe the author refers to the blastocyte stage before telomerase is more active. Clear separation and clarification of these possibilities should be key to understand what is inherited and then what telomere length is extended by telomerase. The sentence has been modified so as to be more clear (Nonetheless, the telomere length of the embryos during the cleavage stages seems to have paternal origin, but the telomere length can be further modified/extended by telomerase activity during the blastocyst stage).
  • Lane 47. Gradually and progressively are synonyms. Just use “telomeres progressively shorten as age progresses, resulting in the activation of a permanent state of cell cycle arrest known as senescence”. The sentence has been modified according to the suggestion of the Reviewer.
  • Lane 53-55. This statement is only valid in condition where telomerase is active, which during early development and to maintain telomeres in stem cells. However, it is inactivated in somatic cells where telomeres shorten in each cell division. Cells don’t adopt the mechanism of telomerase-mediated telomere elongation. Telomerase re-activation in somatic cells occurs as a consequence of activating mutations or chromosomal rearrangements during tumorigenesis. The sentences provided by the Reviewer were added in the text.
  • Lane 70. Aneuploidy in embryos due to telomere shortening. This is interesting to happen in sperm cells, and is closely related with the fact that short unprotected telomeres can fuse with other chromosome free ends (either telomeres or internal sequences) leading to aneuploidy during chromosomal segregation in cancer development. It will be interesting to add this idea. This connection will be interesting to work on to complete the introduction section. The suggested sentences have been added in the text.
  • Lane 81-83. Should say “Telomere undergo progressive shortening following each cell division, until they reach a critical length where senescence is activated”. Shortening is not exactly intensified by age, is intensified by many endogenous and environmental factors which may accumulate as age progress. A reference to the work of Leonard Hayflick should be added. The sentence has been corrected and the suggested Reference has been added (Hayflick 1997)
  • Lane 85-86. Again, I don’t think equilibrium between shortening and lengthening is what is happening in all situations. That could be the case of early development or what is happening in stem cells, however somatic cells follow progressive shortening unless telomeres are damages and the associated repair led to transient lengthening. See your lanes 117-119, that is the general accepted model. Part of the sentence has been deleted and modified so as to be clear by adding more explanation (Therefore, the telomeres of the chromosomes exhibit a dynamic state, where the number of the repeats in the telo-84 meres are depending upon equilibrium or a homeostasis state that controls the shortening 85 and the lengthening procedure during cell cycle. in which the telomerase in germ cells is activated, while in somatic cells it is not activated.
  • Lane 89-92. Telomere length is established during development (which determine the mitotic clock), then is progressive shortening leading to aging. The fact that telomeres shorten with each cell division does not involve that a mechanism to elongate them should be activated. This only happens in cancer cells due to telomerase re-activation that is required to maintain their high proliferative potential. Maybe the authors refer to specific cell lines or systems, but it should be stated. The sentence has been modified according to the suggestion of the Reviewer (Telomere length is established during development, determining the mitotic clock and then is progressively shortened, approximately 50-200 bp in each cycle [15,16], leading to aging.)
  • Lane 93. References for shelterin complex is needed. Authors could add references from Titia de Lange lab. A reference has been added (Schmutz I, de Lange T 2016 Cell Biol).
  • Lane 95. When talking about telomerase, and maybe in other cases, I think the authors should acknowledge its discovery by adding the appropriate reference. Indeed, in this specific case, a nobel prize was awarded to Elizabeth Blackburn, Carol Greider and Jack Szostak for the discovery of how chromosomes are protected by telomeres and the enzyme telomerase. In general, references related to telomere section are missing. The reference suggested by the Reviewer has been added in the reference list (Blackburn EH, Greider CW, Szostak JW. 2006).
  • Lane 99. POT1 is called “Protection of telomeres protein 1”. It has been corrected.
  • Lane 103. The existing sentence has been removed and was replaced by the one suggested by the Reviewer.
  • Lane 110-112. “Generally, it is well established that…”. This type of statements required references. The sentence has been modified and a reference has been added (Honig LS et al 2015)
  • Lane 125. ALT is activated in normal somatic cells. The references provided are referring to ALT mechanism rather than specific activation in normal cells. The references have been removed and replaced by another relative reference (Neumann AA et al 2013)
  • Lane 126-129. Break-induced replication (or Break-induced telomere synthesis) is the mechanism behind telomere elongation by ALT pathway. Homologous recombination repair telomeres but does not lead to elongation. The ALT mechanism has been the focus of intensive research in the last years, the author needs to add appropriate and more recent bibliography. The sentence has been erased (line 129) as suggested by the Reviewer and updated references have been added (Kockler ZW et al 2021; Yang Z et al, 2021)
  • Lane 134. Senescence can actually be bypassed by inactivating mutations in proteins involved in cell cycle regulation such as p53 and RB. This is actually the basis of tumorigenesis in aged people. The sentence provided by the Reviewer has been added in the text (Actually, senescence can be bypassed by inactivating mutations in proteins involved in cell cycle regulation, such as p53 and RB. The above is the basis of tumorigenesis in aged people).
  • Lane 137. “which consequently may influence either telomerase activity or the telomere-shortening”. It is known that eternal factors have an influence, a reference is needed. A reference has been added (Welendorf C et al.2019).
  • Lane 147. Are the authors talking about the length of culturing or also the composition of the culture media? The sentence has been modified so as to be more clear (….the prolonged role of the culture media that are implemented during manipulation of gametes and culture of pre-implantation embryos).
  • Lane 154. “the telomeric repeat-containing RNAs, known as TERRA, have been associated with embryo quality in IVF/ICSI-ET treatments”. It should be clear whether this association is positive or negative. In the sentence has been added the word “positive”.
  • Lane 157. “In the present study, a number of factors have been investigated” What study the authors refers to. From the context, I may guess is reference 36. The sentence has been rephrased (…In that study)
  • Lane 159. It is not clearly documented, actually the authors of reference 36 said more work is needed in that way. I don´t think the statement of a relationship between telomere length and time of culture could be done from a reference where it wasn´t tested. The sentences have been rephrased so as to be more clear (It appears that the telomere length of the neonates, Although not yet documented, it can be hypothesized that telomere length might have been affected by culture embryo extension during the ART process, which epigenetically and indirectly impacts on pregnancy success rates).
  • Lane 163-181. This section long and week on the information that supports it, only detected in Drosophila (it should specifically in Drosophila, rather than particularly in Drosophila). Maybe the authors have some unpublished observations that supports that. It could be either removed completely or the authors could only relocate the idea of RT activity on compensating for telomerase inactivation in the previous section. The section has been modified and replaced in the previous sections as suggested by the Reviewer
  • Lane 184. “sperm DNA should be contributed to the zygote”. Maybe, should be transferred to the zygote. It has been corrected according to the suggestion of the Reviewer.
  • Lane 192. Q-FISH is defined as Quantitative fluorescence in situ hybridization (Q-FISH). It has been corrected
  • Lane 193. “sperm telomere length was approximately 10 to 20 Kb”. What is the variation with age rather than telomerase? Chromosomal reorganizations are involved in telomerase re-activation in some cancer types, could be happening the same in the late phases of spermatogenesis. The sentence has been deleted.
  • Lane 205. “Moreover, it has been documented that telomeres of older men are longer than those of younger individuals”. Because is documented, a reference is needed. The sentence has been modified in order to be more clear and a reference has been added (older paternal age at conception predicts longer TL in offspring, Eisenberg DTA et al. 2019).
  • Lane 210-212. “The question raised is what is the crucial cut-off point of paternal age at which point telomere length is starting to increase and remains to be elucidated taking into consideration the heterogeneity of STLs”. Need to be rephrased for a clearer understanding. The sentence has been modified in order to be more clearer (The question raised is what is the crucial cut-off point of in which paternal age at which point telomere length is starting to increase and remains to be elucidated taking into consideration the heterogeneity of STLs).
  • Lane 233-234. “Shorter STLs seems, therefore, to compromise male infertility, since abnormal protamination has been linked to shorter STLs” It should just say, “Abnormal protamination has been linked to shorter STLs which therefore compromise male fertility”. The sentence has been rephrased according to the suggestion of the Reviewer
  • Lane 243-244. “ROS, which in turn promotes an enhanced percentage of sperm DNA fragmentation”. A reference is needed here. A reference has been added (Asadi A et al. 2021).
  • Lane 246-249. “Interestingly, ROS levels seem to play a crucial role in determining STLs Normal or mild oxidative stress, which could be expressed through the evaluation of ROS levels, is associated with longer STLs in comparison to severe oxidative stress which seems to be related to shorter STLs”. Hard to understand, it should say “Interestingly, while normal and mild oxidative stress is associated to longer STL, severe oxidative stress is related to shorter STL”. The sentence has been replaced as the Reviewer suggested.
  • Lane 252. rs2736100the. Remove “the” and authors should define better this polymorphism. The whole sentence has been replaced and the rs has been defined as suggested by the Reviewer (the TERT rs2736100 was inversely associated with male infertility risk, whereas TEP1 rs1713449 was positively associated with risk of male infertility
  • Lane 257-259. If the data is limited, then there is not a striking association. It should say, “even though data is limiting it seems that telomere length in female gametes follow a different pattern compared to sperm” The sentence has been replaced as suggested by the Reviewer
  • Lane 271-272. “whether truncated telomeres trigger oocyte quality during advanced maternal age, remains to be elucidated”. Authors should re-phrase or elaborate the possibilities on how this may happen. The sentence has been re-phrased according to the suggestion of the reviewer in order to be more clear (It is possible that the advanced reproductive age is the trigger agent for the shortening of the telomeres of the oocytes and truncated telomeres trigger oocyte quality during advanced maternal age).
  • Lane 279-280. “It is, therefore, evident that, ROS levels are crucial and essential for both telomeres lengthening and shortening”. It should be added that “depending on ROS levels, it could lead to either telomere shortening or lengthening”. The sentence has been rephrased according to the suggestion of the Reviewer
  • Lane 311-314. Is this referred to both gametes? The sentence has been modified (However, The correlation between TLs and the methylation – de-methylation cycle of the gametes, including the oocytes has been not remains to be elucidated in an extent.
  • Lane 314. “…seems to present with significant correlation” change to “seems to correlate” It has been changed according to Reviewer.
  • Lane 315-316. Here, reference 66 demonstrate a link between methylation and TL. It should be written what is the link (shortening, lengthening…). Besides that, this sentence seems to contradict another statement in lane 317, where authors said that “the impact of aberrant methylation to TLs cannot be excluded”. Please clarify if the relationship cannot be excluded or was demonstrated but further studies in the context of telomeric chromatin. The sentences have been corrected as the Reviewer suggested (In one study, a link positive correlation between DNA methylation and TLs was demonstrated [66]. It is also known that On the other hand aberrant methylation leads to various disorders, and therefore, the impact of aberrant methylation to TLs cannot be excluded. It is possible that abnormal methylation of the telomeres may result in telomere dysfunction, which subsequently may promote telomere erosion. but further studies are needed in the context of telomeric chromatin. It seems that epigenetic signatures established during the early stages of gametogenesis may contribute more extensively to determining the TL and not that TL defines the fate of the methylation state.
  • Lane 322. “…may raise further concerns”. I think additional studies “will uncover mechanistic clues” The sentence has been modified
  • Lane 333. Reaching the levels instead of resuming the levels. The sentence has been changed
  • Lane 339. “TA remain low throughout the oogenesis process”. In lanes 260-262, the author said the TA is high at early stages and low at the final phase of oogenesis. Please clarify. Part of the sentence has been deleted in order to be more clear (while TA remains low throughout the oogenesis process and varies between initially high and late low levels during oogenesis and the spermatogenesis process)
  • Lane 343. As mentioned above, TL reset theory represent the accepted process to determine telomere length. What is happening before that state during fertilization and the initial stages of embryogenesis represents is obscure and the authors here have a great opportunity to fill some information in. Information was added in lines 325-334.
  • Lane 344-350. References are needed. The sentence has been rephrased (Therefore, it becomes evident that the TLs of the embryos are determined by inherited by the TLs of the gametes. It is very possible that gametes with long telomeres will result in embryos with long telomeres. Following in the same line, embryos with short telomeres might have been originated from gametes presenting with short telomeres)
  • Lane 361-363. “…although the differences in the length of telomeres between oocyte and spermatozoon during fertilization”. It seems incomplete. The sentence has been rephrased (Although the differences in the length of telomeres between oocyte and spermatozoon at the time of fertilization Moreover, it appears that the embryo inherits the longer telomeres)
  • Lane 364. If telomeres are longer during fertilization process, I guess this will be the base for longer telomere extension and life span when telomerase gets activated during blastulation. However, telomere length has a limit which depend on the amount of shelterin protein that can protect the DNA structure. In that case, a protein called TZAP bind to DNA ends and trimmer telomeres to the length that can be protected by the levels of shelterin proteins. Information was added according to the suggestion of the Reviewer (Li JS, Miralles Fusté J 2017)
  • Lane 372-373. “…is well documented”. Do the authors have references for this? (Ruebel ML, Latham KE 2020)
  • Lane 374-375. “…have been observed”. Reference needed. (Blyth U et al. 2021)
  • Lane 377. DNA repairing polymerase? What mechanism do the authors refer to here? Is it a polymerase for HR-mediated repair mechanism? The sentence has been deleted (Whether there is a mechanism that exhibits similar repair properties such as the DNA repairing polymerase, in order to restore telomere lengths during the fertilization process remains to be elucidated)
  • Lane 383. Telomerase-mediated telomere elongation and HR-mediated repair at telomeres are not exclusive mechanisms and can be taking place at the same time, however the first one lead to elongation while the second one lead to telomere length heterogeneity as has been described in ALT cells. Telomere elongation independent of telomerase in ALT-related break-induced replication which may be taking place also (lanes 389-392). However, I’m agreed that further investigation will be needed to elucidate that. We agree with the Reviewer
  • Lane 390. It has to be determined whether… The word indicated has been replaced by the word determined
  • Lane 394. What abnormalities the authors refer to? Length, methylation status…The word abnormality was replaced by the words length of methylation status
  • Lane 397-400. “Additional data…”. References. (References for these sentences are provided in the next section (No 6) as stated
  • Lane 405. “TL has been proposed to correlate with both male and female infertility” At this point in the review, this statement should be clearer because the correlation between age and TL is positive in the case of male and negative in the case of females (also for lane 464). Therefore, the statement should be re-phrased to say that short telomeres correlate with infertility. We agree with the Reviewer. The word “shorter” has been added.
  • Lane 406. First time Assisted reproductive techniques (ART) is defined and acronym is not used. Line 401 (Assisted Reproductive Therapy (ART)). It has been corrected in the whole ms
  • Lane 408. “…abnormal methylation/de-methylation cycle”. Reference missing. This reference should be used for lanes 423-424. (Reference  Anifandis et al. 2015)
  • Lane 415. Is it know the relationship between motility and TL, and how is the variability of TL in a sperm sample population? A reference was added for the relation between STL and sperm motility (Iannuzzi A et al. 2020) and two other references for association between poor sperm quality and shorter TL (Darmishonnejad Z et al. 2020)
  • Lanes 446. “…altered gene expression of the main gene telomere complex…”. It should say “ altered gene expression of shelterin and telomere-associated proteins such as…” It has been changed according to the suggestion of the Reviewer
  • Lane 450. Change “characterize” for “with”. The word characterized has been replaced
  • Lane 452. IUGR not defined (Intrauterine Growth Restriction).
  • Lane 472. PGT-A not defined. (Pre-implantation Genetic Testing for Aneuploidy)

Reviewer 2 Report

Telomere biology is one of the most relevant research areas today, since the correlation between telomere length and aging has been unambiguously shown. A plethora of studies have provided some valuable insights into the role of telomere biology in the pathogenesis of age-related diseases. However, the role of telomeres in gametogenesis and development, as well as the correlation of telomere length with fertility, especially in humans, is much less studied and poorly summarized. Therefore, the submitted MS, in my opinion, is extremely useful for specialists working in the field of developmental biology and reproductive medicine.

Major point

The MS is written logically and fully consistent with the goals and objectives formulated by the authors in the Introduction. After reading the manuscript, no serious questions and comments arose, except for a discussion of telomere length in sperm and oocytes. In lines 63—64, authors point that “… the length of the oocytes’ telomeres, appear to be shorter than in the spermatozoa”, whereas in lines 355—357, they write: “The telomeres of the oocytes are longer than those of the spermatozoa [10]…” I recommend the authors to clarify or correct this discrepancy.

Minor (technical) points

Line 96 – typo: «ribunocleoprotein», correctly ribonucleoprotein

Line 247 – a full point is apparently missing between the words “STLs” and “Normal”

Line 290 – extra capital letter (“Furthermore, Exogenous and…”)

Line 311 – extra punctuation mark (“Except from TLs. another biological and molecular marker…”)

Lines 315—316, 453—461 – font of a different size

Lines 369 and 371 – “di-methylation” – obviously there must be de-methylation

Author Response

We thank the Reviewer for the comments towards improving our Review. The below answers are given to criticism

Major point

The MS is written logically and fully consistent with the goals and objectives formulated by the authors in the Introduction. After reading the manuscript, no serious questions and comments arose, except for a discussion of telomere length in sperm and oocytes. In lines 63—64, authors point that “… the length of the oocytes’ telomeres, appear to be shorter than in the spermatozoa”, whereas in lines 355—357, they write: “The telomeres of the oocytes are longer than those of the spermatozoa [10]…” I recommend the authors to clarify or correct this discrepancy.

We thank the reviewer for this indication. The first sentence has been corrected

Minor (technical) points

All minor points have been addressed according to the suggestions of the Reviewer

Line 96 – typo: «ribunocleoprotein», correctly ribonucleoprotein

Line 247 – a full point is apparently missing between the words “STLs” and “Normal”

Line 290 – extra capital letter (“Furthermore, Exogenous and…”)

Line 311 – extra punctuation mark (“Except from TLs. another biological and molecular marker…”)

Lines 315—316, 453—461 – font of a different size

Lines 369 and 371 – “di-methylation” – obviously there must be de-methylation

Round 2

Reviewer 1 Report

In this revised version, George Anifandis and colleagues have made an effort to include all comments/suggestions. However, this reviewer thinks that some of the suggestions were just included but not incorporated in the text in a way that fit with it. Overall, I think this revised version is acceptable in the present form with minor additional changes as described below. These final changes, together with the language and formatting improvements from the Journal of Developmental Biology will in my opinion further improve the quality of this revision.

Specific Comments:

Lane 20-22: Better to simply say: “Telomeres promote genome integrity by protecting chromosome ends from the activation of the DNA damage response, and protect chromosomes from the loss of coding sequences due to the end replication problem”

Lane 23: Gradually and progressively are still synonyms, please remove one of the two.

Lane 73-77 (red text): Better to simply say. “In this regard, this aneuploidy is also frequently detected during cancer development and initiated by critically short telomeres” 

Lane 89: quotation mark forgotten to remove.

Lane 92: Telomere length (TL) was defined before.

Lane 87-96. These sentences in red were explanations to the authors of the mechanims, not sentences to include in the text as it. Better replace from 87 to 96 for something like this:

“Telomere length is established in germ cells during development by the activity of telomerase, determining the mitotic clock of each cell (16). In somatic cells, telomerase is inactivated leading to progressive telomere shortening, approximately 50-200 bp following in each cell division, until they reach a critical length where senescence is activated. Besides progressive telomere shortening due to the end replication problem, telomere shortening is intensified by many genetic, environmental and life style factors (14, 15, 16), therefore, accounting for the high variability of telomere length observed in humans”

Lane 117-120 and lanes 140-144. Both are talking about endogenous and exogenous factors, please fuse both together to not being over repetitive in the paragraph.

Lane 120-123: “Additionally, an explanation…not activated in somatic cells”. Remove this sentence as the idea for germ cells and somatic cells was included in the first paragraph of that section. It will be better to focus this paragraph on the other factors.

Lane 133-134: “The mechanism behind telomere elongation by ALT is known as Break-induced replication (or Break-induced telomere synthesis)” This mechanism in humans has been determined by work at Roger Greenberg (“Break-induced telomere synthesis underlies alternative telomere maintenance”) and Hilda Pickett labs (“BLM and SLX4 play opposing roles in recombination-dependent replication at human telomeres”). Please, include at least these references.

Lanes 135-140: The sentence “…the senescence-aging phenomenon cannot be bypassed” does not make sense with the next “Actually, senescence can be bypassed…). Better say here “the senescence-aging phenomenon cannot be bypassed unless inactivating mutations accumulate in proteins involved in cell cycle regulation, such as p53 and RB. The latter represents the basis of tumorigenesis in aged people.

Lane 149: BMI defined for the first time.

Lane 169: Start the paragraph with “Finally” as it is the last argument in this section. Maybe the author could consider add something else to the title of this section rather than just “The role of telomeres”. The role of telomeres in embryos. The role of telomeres in gametes or embryos…

Lane 267: “…truncated telomeres trigger oocyte quality…” The word you may want is “...truncated telomeres decrease oocyte quality…”

Author Response

We thank the Reviewer for the constructive criticism on our review. All minor points have been corrected and the suggested references have been added according to the suggestions of the Reviewer

Lane 20-22: Better to simply say: “Telomeres promote genome integrity by protecting chromosome ends from the activation of the DNA damage response, and protect chromosomes from the loss of coding sequences due to the end replication problem”

Lane 23: Gradually and progressively are still synonyms, please remove one of the two.

Lane 73-77 (red text): Better to simply say. “In this regard, this aneuploidy is also frequently detected during cancer development and initiated by critically short telomeres” 

Lane 89: quotation mark forgotten to remove.

Lane 92: Telomere length (TL) was defined before.

Lane 87-96. These sentences in red were explanations to the authors of the mechanims, not sentences to include in the text as it. Better replace from 87 to 96 for something like this:

“Telomere length is established in germ cells during development by the activity of telomerase, determining the mitotic clock of each cell (16). In somatic cells, telomerase is inactivated leading to progressive telomere shortening, approximately 50-200 bp following in each cell division, until they reach a critical length where senescence is activated. Besides progressive telomere shortening due to the end replication problem, telomere shortening is intensified by many genetic, environmental and life style factors (14, 15, 16), therefore, accounting for the high variability of telomere length observed in humans”

Lane 117-120 and lanes 140-144. Both are talking about endogenous and exogenous factors, please fuse both together to not being over repetitive in the paragraph.

Lane 120-123: “Additionally, an explanation…not activated in somatic cells”. Remove this sentence as the idea for germ cells and somatic cells was included in the first paragraph of that section. It will be better to focus this paragraph on the other factors.

Lane 133-134: “The mechanism behind telomere elongation by ALT is known as Break-induced replication (or Break-induced telomere synthesis)” This mechanism in humans has been determined by work at Roger Greenberg (“Break-induced telomere synthesis underlies alternative telomere maintenance”) and Hilda Pickett labs (“BLM and SLX4 play opposing roles in recombination-dependent replication at human telomeres”). Please, include at least these references.

Lanes 135-140: The sentence “…the senescence-aging phenomenon cannot be bypassed” does not make sense with the next “Actually, senescence can be bypassed…). Better say here “the senescence-aging phenomenon cannot be bypassed unless inactivating mutations accumulate in proteins involved in cell cycle regulation, such as p53 and RB. The latter represents the basis of tumorigenesis in aged people.

Lane 149: BMI defined for the first time.

Lane 169: Start the paragraph with “Finally” as it is the last argument in this section. Maybe the author could consider add something else to the title of this section rather than just “The role of telomeres”. The role of telomeres in embryos. The role of telomeres in gametes or embryos…

Lane 267: “…truncated telomeres trigger oocyte quality…” The word you may want is “...truncated telomeres decrease oocyte quality…”